# The Genetic Susceptibility to Psoriasis and the Relationship of Linked Genes to Our Treatment Options

**DOI:** 10.3390/ijms241512310

**Published:** 2023-08-01

**Authors:** Heli A. Patel, Rishab R. Revankar, Sofia T. Pedroza, Shaveonte Graham, Steven R. Feldman

**Affiliations:** 1Center for Dermatology Research, Wake Forest School of Medicine, Winston-Salem, NC 27104, USA; 2The Icahn School of Medicine at Mount Sinai, New York, NY 10029, USA; 3Baylor College of Medicine, Houston, TX 77030, USA; 4Wright State University Boonshoft School of Medicine, Fairborn, OH 45435, USA; 5Department of Dermatology, Wake Forest School of Medicine, Winston-Salem, NC 27104, USA

**Keywords:** psoriasis, genetics, biologics, cytokines, interleukins, small molecules, JAK-STAT

## Abstract

Understanding the factors creating genetic susceptibility in psoriasis may provide a basis for improving targeted treatment strategies. In this review, we discuss the genes linked to the pathogenesis of psoriasis and their relationship to the available treatment options. To identify the relevant genetic markers and treatments, we searched PubMed, Google Scholar, MEDLINE, and Web of Science with keywords, including *genetic susceptibility to psoriasis*, *genetics and psoriasis*, *psoriasis treatments*, and *biologics treatments in psoriasis*. The articles in English from database inception to 1/1/23 were included. Case reports and series were excluded. Gene variant forms commonly implicated in the pathogenesis of psoriasis include those encoding for interleukins, interferons, and other mediators involved in inflammatory pathways, such as JAK/STAT, and NF-κB. Several of the treatments for psoriasis (for example IL23 and TYK2 inhibitors) target the products of genes linked to psoriasis. Multiple genes are linked to the pathogenesis of psoriasis. This understanding may provide an avenue for the development of new psoriasis treatment strategies and for more effective, safer treatment outcomes.

## 1. Introduction

Psoriasis is a chronic and recurrent inflammatory skin condition characterized by erythematous scaly plaques that most commonly appear on the scalp, elbows, and knees. In addition to cutaneous manifestations, there are also systemic manifestations of psoriasis that include psoriatic arthritis, cardiovascular disease, diabetes, depression, anxiety, certain malignancies, and obesity [1,2,3]. Psoriasis affects 2–3% of the population in the United States and has a serious negative impact on quality of life, as current treatments do not serve to cure but rather to alleviate symptoms [1,2,4,5,6,7,8]. This results in high and often insurmountable healthcare costs for families. Psoriasis has a peak occurrence in both males and females between 20–30 years and 60–70 years of age [1]. There are multiple subtypes of psoriasis, of which the most common is psoriasis vulgaris, also known as plaque psoriasis [2]. Other less common subtypes include inverse or flexural psoriasis, guttate psoriasis, pustular psoriasis, and erythrodermic psoriasis [1,2,7]. The etiology of psoriasis consists of multifactorial components including both genetic and environmental factors [1,4,9,10]. These environmental factors can include chronic infections, stress, low humidity, drugs, smoking, and obesity [1,4,7]. Certain infections such as streptococcal infections, medications, alcohol abuse, and physical trauma may also trigger or worsen existing psoriasis [11].

The role of genetics in disease pathophysiology is being increasingly explored as an approach to targeted treatment. Psoriasis has a significant familial genetic correlation, as more than 20% of patients report family history [9]. Additionally, disease concordance is higher in monozygotic compared to dizygotic twins [3,4,9,10]. There are many genes found to play a role in disease development, and over 80 susceptibility loci were identified [1,6,9]. For example, certain loci identified allow for the nuclear factor kappa-light-chain-enhancer of activated B cells (NF-κB) signal transduction cascade, which is thought to play a role in the pathogenesis of psoriasis. Cytokines, important mediators in inflammatory responses, were linked to psoriasis, such as IL-1, IL-17, and IL-22 [11]. There are genes that are strongly associated with a higher risk of developing psoriasis, such as HLA-C*06:02 [1,11]. Targeting the specific genes that underlie psoriasis could be an effective and safe approach to managing this disease and may provide the foundation for future precision medicine techniques. In this review, we compared the genes linked to psoriasis to available treatment options in order to identify treatments that specifically target the products of genes that cause psoriasis.

## 2. Discussion

The genes associated with psoriasis may be separated into categories that reflect the major immunological mediator and pathway that they influence. These categories act as a system to generate immunological responses; thus, their corresponding genetic mediators have the potential to serve as targets for psoriasis treatment.

### 2.1. Interleukins (IL)

Interleukins are cytokines that play an essential role in the differentiation, proliferation, migration, adhesion, and maturation of immune cells. They can be both pro-inflammatory and anti-inflammatory. By binding to interleukin receptors, interleukins trigger a cascade of reactions that constitute the immune system [12]. The pro-inflammatory mediators associated with psoriasis are IL1B, IL2, IL6, IL12, IL15, IL17, IL18, and IL20. Specifically, IL1B encodes a cytokine produced by activated macrophages and plays an important role in the inflammatory response. IL2 encodes a cytokine important for the proliferation of T and B lymphocytes. IL6 encodes a cytokine that induces inflammatory responses through IL6Ra and the maturation of B cells. IL12 induces Th1 cells to produce IFN-γ. IL15 encodes a cytokine that regulates T-cell and natural killer activation and proliferation and induces the activation of janus kinases (JAK), as well as the phosphorylation and activation of STAT3, STAT5, and STAT6 proteins. IL17 is produced due to IL23 activation of Th17 cells. IL17 is a key mediator of autoimmunity. IL18 stimulates the production of IFN-γ in Th1 cells. IL20 encodes a cytokine structurally related to IL10 and transduces its signal through STAT3 in keratinocytes [4].

The anti-inflammatory genetic mediators associated with psoriasis are IL1RN, IL4, IL10, IL13, and IL19. IL1RN inhibits IL1 and modulates the immune and inflammatory responses. IL4 is a pleiotropic cytokine involved in the modulation of Th2 immune responses. Its receptor also binds IL13, which may contribute to many overlapping functions of this cytokine and IL13. IL10 encodes a cytokine produced by monocytes and lymphocytes that downregulates the expression of Th1 cytokines and blocks NF-kB activity. It further enhances B-cell survival, proliferation, and antibody production and regulates the JAK-STAT signaling pathway. IL13 encodes a cytokine produced by activated Th2 that is involved in the maturation and differentiation of B cells and downregulates macrophage activity and inhibits the production of proinflammatory cytokines and chemokines [4]. IL19 is a member of the IL10 cytokine subfamily with a role in inflammatory responses.

Several genetic variants among ILs are associated with psoriasis development and progressions. The rs2275913 single nucleotide polymorphism (SNP) in the IL17A gene, featuring a substitution of guanine (G) with adenine (A), was linked to an increased risk of psoriasis by enhancing IL17A production and promoting inflammation. Furthermore, the rs3212227 SNP in the IL12B gene, characterized by a substitution of thymine (T) with cytosine (C), is associated with psoriasis susceptibility by upregulating IL12B expression, which contributes to the dysregulated immune response observed in psoriasis. Additionally, the rs1800872 SNP in the IL10 gene, featuring the G allele, was associated with an elevated risk of psoriasis by downregulating the production of IL10, an anti-inflammatory cytokine involved in immune regulation. These genetic variations provide insight into the role of ILs in psoriasis pathogenesis [3,4,9,10]. However, further research is needed to fully understand their functional implications and their precise contribution to the disease course.

### 2.2. Interleukin Receptors (ILR)

Interleukin receptors (ILRs) are high-affinity receptor proteins that bind interleukins to initiate the immune cascade reactions associated with each type of interleukin molecule [12]. The ILR-encoding genes associated with psoriasis are IL20RA, IL28RA, and IL36RN. IL20RA encodes a receptor for IL20, a cytokine that may be involved in epidermal function. IL28RA encodes a receptor complex that interacts with IL28A, IL28B, and IL29 [4]. The expression of these cytokines can be induced by viral infections. IL36RN encodes the IL36 receptor antagonist protein and the loss of function mutations leads to dysregulated IL36R signaling, which is linked to disease onset [3,4].

Specific variants of ILRs are associated with psoriasis. This includes the rs11209026 variant of IL23R, an SNP involving a substitution of adenine (A) with guanine (G). Similarly, an SNP associated with IL20R (rs17728338) featuring the G allele was associated with an increased risk of psoriasis. Furthermore, the C allele present in the rs7631529 SNP found in the IL36RN gene was correlated with an increased risk of psoriasis. The common understanding among these SNP variants is that they predispose dysregulated interleukin signaling and thereby increase inflammatory mechanisms associated with disease pathophysiology [3,4,9,10,12]. Further research is necessary to better understand the implications of the underlying mechanisms of these genetic variations in psoriasis development.

### 2.3. Human Leukocyte Antigen (HLA)

Human leukocyte antigens (HLA) are genes in major histocompatibility complexes (MHCs) that help the body differentiate between proteins that are self and non-self. These antigens are highly variable, especially class I HLA antigens: HLA-A, HLA-B, and HLA-C. These antigens are located on all nucleated cells and function to present identifying proteins to cytotoxic T-cells for subsequent immune reactions [13]. The two genes relevant to this category associated with psoriasis are HLA-C and ERAP1. HLA-C plays a central role in the immune system by presenting peptides derived from endoplasmic reticulum lumen. ERAP1 encodes an aminopeptidase involved in trimming HLA class I-binding precursors [4].

HLA-C*6:02 is the major psoriasis risk allele that is located in psoriasis susceptibility locus 1 (PSORS1) within the short arm of chromosome 6p21 [10]. The PSORS1 locus encodes genes that play a role in antigen presentation to CD8+ T lymphocytes. Among the HLA types, HLA-C*6:02 has emerged as the strongest genetic risk factor for psoriasis, with the rs10484554 SNP within the gene showing a significant association. Additionally, the rs4349859 SNP in the HLA-B gene, specifically associated with HLA-B27, was implicated in an increased risk of psoriasis and psoriatic arthritis. Another risk allele, rs11652075, found in HLA-B13, was linked to an increased susceptibility to psoriasis. Furthermore, the rs660895 SNP within the HLA-DRB1 gene has shown an association with an increased risk of psoriasis, particularly in early-onset cases. Conversely, the rs3099844 SNP in HLA-B57 was associated with a decreased risk of psoriasis. These SNP variants in HLA genes highlight the important role of HLA-mediated immune responses in psoriasis pathogenesis, although further research is required to fully elucidate the precise mechanisms underlying these associations [3,4,9,10,13].

### 2.4. Nuclear Factor Kappa B (NF-kB)

Nuclear factor kappa B (NF-κB) is a transcription factor activated by two major signaling pathways: the classical (canonical) signaling pathway and the alternative pathway. NF-κB is known to regulate multiple aspects of adaptive and innate immune responses, as well as inflammation. For example, NF-κB helps induce pro-inflammatory cytokines and regulate T-cell activation and differentiation. It was also implicated in many other inflammatory diseases such as multiple sclerosis, rheumatoid arthritis, and asthma [14]. The genes associated with psoriasis that encode regulators of the NF-kB pathway include TNFAIP3, TNFRSF1B, TNIP1, TRAF1IP2, and NF-κ-BIA. TNFAIP3 is induced by TNF, which inhibits NF-κB activation and TNF-mediated apoptosis and is involved in cytokine-mediated immune and inflammatory responses. TNFRSF1B is a TNF-α receptor that mediates the recruitment of antiapoptotic proteins. TNIP1 encodes TNFAIP3 interacting protein 1, which plays a role in the regulation of NF-κB activation. TRAF1IP2 encodes a protein that interacts with TRAF proteins and plays a central role in innate immunity in response to pathogens, inflammatory signals, and stress. NF-kBIA encodes a member of the NF-κB inhibitor family, which interacts with REL dimers to inhibit NF-κB/REL complexes involved in inflammatory responses [4].

Several SNPs in the genes involved in the TNF signaling pathway were implicated in psoriasis susceptibility. The rs2230926 SNP in the TNFAIP3 gene, characterized by the G allele, was associated with an increased risk of psoriasis. This variant affects the regulation of TNF-alpha-induced signaling pathways, contributing to the dysregulation of the immune responses observed in psoriasis. Similarly, the rs1061624 SNP in the TNFRSF1B gene, featuring the T allele, was linked to psoriasis susceptibility, potentially disrupting the TNF receptor signaling and promoting chronic inflammation. Additionally, the rs7708392 SNP in the TNIP1 gene was associated with psoriasis, suggesting a role in modulating NF-κB activity and immune responses. The rs929386 SNP in the TRAF1IP2 gene was also implicated in psoriasis susceptibility, potentially affecting the NF-κB signaling pathway. Furthermore, the rs696 SNP in the NF-κBIA gene, characterized by the A allele, was linked to an increased risk of psoriasis, potentially altering the negative regulation of NF-κB and promoting inflammation. These SNP variants shed light on the involvement of TNF signaling and NF-κB pathway dysregulation in the pathogenesis of psoriasis [3,4,9,10,14]. However, further research is needed to fully elucidate the extent of their role in disease development and progression.

### 2.5. Interferons (IFNs)

Interferons are cytokine mediators that activate the immune system as part of an inflammatory response to pathogens, damaged cells, or irritants. Type I IFNs are released by virus-infected cells, while type II IFNs, such as IFN-γ, are released predominantly by T-cells. Their role includes promoting cytokine production and amplifying antigen presentation to immune cells [15,16]. The genes associated with psoriasis that encode IFNs and IFN mediators are IFN-γ and IFIH1. IFN-γ encodes a soluble cytokine with antiviral, immunoregulatory, and antitumor properties and is a potent activator of phages. IFIH1 encodes a protein that mediates the induction of the IFN response to viral RNA [4].

For IFN-γ, the rs2430561 SNP is associated with the increased risk of developing psoriasis. This variant involves a substitution of adenine (A) with guanine (G) and is thought to affect the production or activity of IFN-γ, a pro-inflammatory cytokine involved in immune responses. In the case of IFIH1, the rs1990760 SNP was linked to psoriasis susceptibility. This variant is characterized by the T allele and is believed to upregulate the IFIH1-mediated immune cascade [9,10,15,16].

### 2.6. The Janus Kinase (JAK)-Signal Transducer and Activator of Transcription (STAT) Pathway

The Janus kinase (JAK)-signal transducer and activator of transcription (STAT) pathway exists downstream of cytokine receptor binding. Upon cytokine binding, JAK proteins are activated, which, in turn, activates STAT proteins via tyrosine phosphorylation [17]. The genes associated with psoriasis that encode players in the JAK-STAT pathway are STAT4 and TYK2. STAT4, like all proteins in the STAT family, is phosphorylated and translocated to the cell nucleus, where it acts as a transcription activator in response to cytokines. STAT transduces IL12, IL23, and IFN type I signals in T lymphocytes and regulates the differentiation of Th cells. TYK2 encodes a member of the JAK protein family that promulgates cytokine signals by phosphorylating receptor subunits. A component of the IFN I and II signaling pathways may play a role in antiviral immunity [4].

SNP variants in several members of the JAK, STAT, and TYK gene families were implicated in psoriasis. For JAK2, the rs34536443 SNP was associated with psoriasis susceptibility. This variant involves the C allele and is thought to influence JAK2 signaling, which plays a role in immune cell activation. In the case of STAT3, the rs744166 SNP was linked to increased risk of psoriasis. This variant is characterized by the G allele and is believed to affect STAT3 function, a transcription factor involved in immune responses. Additionally, the rs34536443 SNP in TYK2 was associated with psoriasis susceptibility. This variant involves the G allele and is thought to impact TYK2 signaling, which is involved in cytokine signaling pathways. Additional research may aid in better understanding the functional consequences and detailed molecular mechanisms by which these SNP variants influence the manifestation and progression of psoriasis [9,10,17].

### 2.7. Other

Other psoriasis-linked genes include APOE, CTLA4, DEFB4, GBP6, LCE, MCP1, VDR, and ZNF313 (Table 1). APOE was previously linked to hypercholesterolemia and hypertriglyceridemia, both of which are correlated with psoriasis. APOE gene polymorphisms were further implicated in the pathogenesis of psoriasis, as APOE plays a role in the proliferation of T lymphocytes and protects against some infections in patients with psoriasis [4,18]. CTLA4 encodes a protein that inhibits T cells and has an inverse correlation with psoriasis severity, as psoriasis is characterized by increased T-cell infiltration [4,14]. DEFB4 is a member of a family of microbicidal, cytotoxic peptides made by neutrophils [4]. It encodes human β-defensin 2, which has an essential role in skin inflammation [19]. GBP6 is an interferon that induces GBP, which hydrolyzes GTP to both GDP and GMP [4]. LCE encodes a protein that plays a role in skin barrier function, with its expression being largely restricted to the epidermis. Certain subtypes of LCE have anti-microbial properties that may play a role in the innate immunity of the skin, especially the stratum corneum [20]. In particular, LCE3B and LCE3C deletions are highly implicated as a psoriasis risk factor due to interactions with the HLA-C*06 gene. LCE3B and LCE3B are located in the epidermal differentiation complex on chromosome 1 [20]. Deletion in LCE3B and LCE3C also causes increased expression of the upstream LCEA gene, which alters the skin balance of self-peptides and the microbiome. MCP1 encodes a cytokine characterized by two cysteines separated by a single amino acid that displays chemotactic activity for monocytes and basophils [4]. MCP1 was indicated as a potential inflammatory marker in determining psoriasis severity, as it is locally and systemically augmented in psoriatic patients [21]. VDR encodes the nuclear hormone receptor for vitamin D3, which regulates immune response pathways [4]. These receptors play a role in epidermal barrier function, antimicrobial processes, and inflammation in healthy individuals, as well as in the inflammation seen in patients with autoimmune diseases such as rheumatoid arthritis, systemic lupus erythematosus (SLE), and psoriasis [22]. ZNF313 encodes a protein that is involved in T-cell activation [4]. It is abundantly expressed in the skin, as well as T-lymphocytes and dendritic cells for immune function [23].

### 2.8. Specific Drug Targets

The discovery of gene associations enabled the development of safe and effective drugs that directly target genes responsible for psoriasis, such as biologics antagonizing tumor necrosis factor (TNF), the p40 subunit of both interleukin (IL)12 and IL23, and the p19 subunit of IL-23 (IL23p19) and IL17 (Table 2) [5]. These drugs may be well tolerated and improve the PASI-75 (Psoriasis Area and Severity Index reduction >/= 75%) score at week 12 [4]. Treatments such as ustekinumab and briakinumab target the p40 subunit of IL12/23 and are highly effective, underscoring the role of IL12 and IL23 in the pathophysiology of psoriasis [1,3,5,6]. In the case of ustekinumab, the mechanism of action prevents the interaction between IL12/23 and IL12-β-3, not only preventing downstream cascades but also gene activation and cytokine production. IL12 is primarily produced by dendritic and phagocytic cells in response to microbial infection to activate NK cells and T-lymphocytes. IL12 also mediates Th1 cells, which play a vital role in cell-mediated immunity and macrophage infiltration. Furthermore, IL23 is crucial to Th17 differentiation. Th17 cells produce a variety of pro-inflammatory cytokines in humans, such as the IL17 necessary for neutrophil recruitment, IL21 for B cell function, and IL23 for additional Th17 differentiation [24]. Therefore, both ustekinumab and briakinumab block the inflammatory cascade triggered by Th1 and Th17 lymphocytes. Dysregulation in both of these inflammatory processes is linked to psoriasis, as well as other immune disorders such as rheumatoid arthritis and multiple sclerosis [1,6,24,25].

Treatments such as guselkumab, tildrakizumab, and risankizumab target the p19 subunit of IL23 [1,3,5,6]. This subsequently blocks the release of other cytokines, as IL23 facilitates Th17 differentiation. Given the role of Th17 in triggering pro-inflammatory cascades such as that of IL17 and IL21, the aforementioned biologics serve to halt the immune dysregulation contributing to disease pathophysiology. Notably, these drugs have no affinity for IL12 and, thus, allow the IL12/Th1 axis to remain intact, as IL12 initiates the differentiation of naive CD4+ T cells to Th1 cells necessary for cell-mediated immunity. This particular characteristic of these three drugs may be important, given that the literature suggests that dysregulated IL23 may play a larger role than IL12 overproduction in psoriasis pathogenesis. Notably, IL23p19 mRNA is elevated in skin samples from psoriatic patients [26]. IL23 is also involved in IL-C3 cell activity, which causes epidermal thickening in psoriasis. Intradermal injections of IL23 were shown to cause psoriatic skin lesions to develop in mice models [27,28]. Guselkumab, tildrakizumab, and risankizumab selectively inhibit IL23, unlike ustekinumab, which is a non-selective IL23 inhibitor.

Additionally, infliximab, etanercept, and adalimumab target TNF-α, which induces the production of proinflammatory cytokines and mediators such as IL1 and IL6 and helps recruit immune cells to sites of inflammation [1,5,6,29]. The inhibition of TNF-α leads to a reduction in inflammation [1,6]. Through its receptor, TNFR1, TNF can activate intracellular pathways that dictate immune pathways, such as that of NF-κB and mitogen-activated protein kinase (MAPK). The anti-inflammatory outcomes observed with infliximab may also be due to the drug’s enhancement of the regulatory T-cell response, which is immunosuppressive and, thus, useful in the treatment of psoriasis. Blocking TNF by monoclonal antibodies was also indicated in other immune disorders such as inflammatory bowel disease (IBD) and hidradenitis suppurativa. Infliximab was shown to decrease epithelial cell apoptosis in patients with IBD [29], which may be similar to the mechanism of action in psoriasis treatment.

Other drugs target TRAF3IP2, such as secukinumab, ixekizumab, and brodalumab [1,5]. These drugs work by blocking IL17 and IL17R, which play key roles in the recruitment of neutrophils in normal immune function, as well as in the pathogenesis of psoriasis [1]. Secukinumab selectively inhibits IL17A, further reducing its downstream targets such as β-defensin 2, a peptide that induces inflammatory mediators, and its upstream mediator IL23. This mechanism was associated with psoriatic plaque resolution [30]. Ixekizumab and brodalumab were proposed to work similarly.

Apremilast inhibits the enzyme phosphodiesterase-4 (PDE4), which hydrolyzes cyclic adenosine monophosphate (cAMP), thereby affecting the downstream signaling responsible for inflammation and immunity [31]. More specifically, apremilast indirectly amplifies levels of cAMP, which, in turn, activates protein kinase A and serves to decrease levels of TNF-α, IL23, and IFN-γ while increasing IL10 [2,3]. Apremilast was shown to reverse and reduce clinical pathophysiology and symptoms seen in psoriasis, as well as psoriatic arthritis. This is, therefore, an example of a treatment that treats psoriasis by restoring the balance of pro-inflammatory and anti-inflammatory mediators rather than targeting one specific component of inflammation [31,32]. 

Spesolimab targets IL36RN, which works by blocking human IL36R signaling [33]. The IL36 pathway is thought to be heavily implicated in the pathogenesis of generalized pustular psoriasis (GPP), although there are uncertainties in the exact mechanism by which IL36 plays a role. Generalized pustular psoriasis is characterized by superficial sterile pustules on an erythematous base with hyperkeratosis and diffuse dermal mononuclear and inflammatory infiltrates on histopathology. A loss of function in the LF36RN gene was implicated in GPP [34]. This differs from plaque psoriasis, which has appeared to rely more heavily on IL17, indicating other routes of treatment depending on the psoriasis subtype [35,36].

Finally, the newest approved drug for psoriasis, deucravactinib, functions to inhibit tyrosine kinase 2 (TYK2), a member of the JAK family. This leads to the inhibition of the IL12, IL23, and IFN type 1 pathway [1,6,37], thus, disrupting the IL23/IL17 axis, all of which are shown to contribute to psoriasis pathogenesis [38].

Other drugs may also indirectly target genes associated with psoriasis (Table 3). These include IL12/IL23 inhibitors that also target ERAP1; TNF-α inhibitors that also target HLA-C and TNFAIP3; methotrexate (MTX), which also targets SMG6, IMMT, UPK1A; adalimumab, which also targets IL1B; and tofacitinib, which inhibits JAK signaling, thus, inhibiting STAT [4]. 

## 3. Limitations

To identify the genes and treatments linked to psoriasis, a literature search utilizing PubMed, Google Scholar, MEDLINE, and Web of Science was conducted with a keyword search for genetic susceptibility to psoriasis, genetics, psoriasis, psoriasis treatments, biologics, and treatments linked to psoriasis. A total of 180 articles were screened by two independent reviewers. The articles that specifically discussed genes and treatment targeting gene products implicated in psoriasis were selected, for a total of thirty-seven articles. All other articles were excluded. Additionally, we screened the reference sections of all included studies. The results included narrative reviews, systematic reviews, editorial letters, and meta-analyses. In this literature review, we identify genes associated with the inflammatory mediators present in psoriasis and classify existing treatment options as (1) treatments that directly target a gene product and (2) treatments that target factors secondarily related to a gene product.

This review has limitations that should be acknowledged. First, the lack of standardized approaches for studying the genetic variants associated with psoriasis complicates the comparison of findings across different studies. Furthermore, the functional implications of the identified genetic variants and their precise mechanisms of action remain unclear, as collecting prospective data would be challenging. Finally, while this review discusses potential gene targets for therapy, it is important to recognize that translating genetic findings into effective treatments can often be complex. These limitations underscore the need for more robust and standardized methodologies in genetic studies of psoriasis.

## 4. Conclusions

Psoriasis is a multifaceted inflammatory skin condition influenced by various genetic factors. Understanding these genetic components is crucial for developing targeted and personalized treatments that may improve the patients’ quality of life. This review has categorized the genes involved in psoriasis based on their impact on immunological mediators and pathways.

Interleukins (ILs), human leukocyte antigens (HLAs), nuclear factor kappa B (NF-κB) pathway genes, interferons (IFNs), and Janus kinase (JAK)-signal transducer and activator of transcription (STAT) pathway genes were all implicated in psoriasis susceptibility and pathogenesis. Individuals with psoriasis may be genetically susceptible due to the immune dysregulation that occurs in the setting of specific, high-risk alleles. By targeting these genes and their related products, novel treatment strategies can be developed to directly inhibit the disease progression.

Existing therapies, such as biologics that target specific ILs or receptors, have shown promise in managing psoriasis. However, further research is needed to optimize use, determine long-term safety profiles, and minimize potential adverse effects. Indirectly targeting factors associated with gene products, such as modulating the NF-κB pathway activity or inhibiting JAK-STAT signaling, also hold potential for future therapeutic interventions.

The advancements in precision medicine and genomics have opened new avenues for personalized treatment approaches. By identifying specific gene targets, healthcare providers can tailor therapies to individual patients, leading to improved treatment outcomes and a higher quality of life. Additionally, exploring the shared mechanisms between psoriasis and other inflammatory diseases may uncover novel treatment strategies that can benefit a broader range of patients.

In summary, this review highlights the importance of genetic factors in psoriasis and their potential as treatment targets. Continued research and collaboration in this field will be instrumental in transforming the lives of psoriasis patients and advancing our understanding of inflammatory skin conditions.

## Figures and Tables

**Table 1 ijms-24-12310-t001:** Genetic mediators associated with psoriasis.

Type		Gene(s)
Interleukins	Pro-inflammatory	IL1B, IL2, IL6, IL15, IL17, IL18, IL20, IL23
	Anti-inflammatory	IL1RN, IL4, IL10, IL13, IL19
Interleukin receptor proteins		IL20RA, IL28RA, IL36RN
Mediators of HLA class I function		HLA-C, ERAP-1
NF-kB pathway regulators		TNFAIP3, TNFRsF1B, TNIP1, TRAF1IP2, NFkBIA
Interferons and mediators		IFN-gamma, IFIH1
JAK/STAT pathway		STAT3, STAT4, JAK2, TYK2
Other		APOE, CTLA4, DEFB4, GBP6, LCE, MCP1, VDR, ZNF313

**Table 2 ijms-24-12310-t002:** The genes linked to psoriasis and their associated direct treatments.

Target Gene	Direct Treatments
IL12B (12p40)	Ustekinumab Briakinumab
IL23R (23p19)	Guselkumab Tildrakizumab Risankizumab BI-655066
TYK2	Deucravacitnib
TRAF3IP2	Secukinumab Ixekizumab Brodalumab
IL36RN	Spesolimab
IL10	rHIL-10
TNF-α	Infliximab Etanercept Adalimumab
cAMP	Apremilast

**Table 3 ijms-24-12310-t003:** The genes and their associated indirect treatments.

Target Gene	Indirect Treatments
ERAP1	Anti-IL12/23
HLA-C	Anti-TNF-α
SMG6, IMMT, UPK1A	Methotrexate
IL1-β	Adalimumab
STAT	Tofacitinib
TNFAIP3	Anti-TNF-α

## Data Availability

No new data were created or analyzed in this study. Data sharing is not applicable to this article.

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
