# Peer review of "The Genetic Susceptibility to Psoriasis and the Relationship of Linked Genes to Our Treatment Options"

_ijms, 2023, doi:10.3390/ijms241512310_

Round 1

Reviewer 1 Report

The paper is a review on genes and their products involved in the pathogenesis of psoriasis.

The article is very interesting; authors make a list of the major genes associated with psoriasis and describe precisely their role in inflammation. The results are well tabulated and clarified in easy-to-understand tables.

All in all, this is a conceptually simple but very useful work that brings together in one article the genes involved in psoriasis, gives an exhaustive list of them, and analyzes the effects of their products. This creates a bridge between the genetic basis and the development of disease. Not only that, new drug therapies to their specific targets are collected and addressed, illustrating how each gene product can be inhibited by a specific drug.

Under a scientific and specific point of view, the paper provides a useful list of the major genetic causes or associations of psoriatic disease. Linking these genes to their target therapies, the authors give many specific therapeutic solutions related to the main genes involved in psoriasis development. In this way, they open new perspectives to a personalised and precision medicine. 

English is very clear.

I would suggest to the authors to explain how they have searched and selected articles useful for review, indicating the number of those found, selected and discarded, not only in the Abstract but also in the rest of the article. Secondly, I would suggest putting the text that comes after paragraph 2.7 and Table 1 into a new paragraph, 2.8, titled, for example, "Specific Target Drugs".

Overall it is a good work, that we can read with curiosity and interest.

Round 2

Reviewer 2 Report

The authors adequately revised the manuscript and replied to the comments.